# Evaluation of the Emulsifying Property and Oxidative Stability of Myofibrillar Protein-Diacylglycerol Emulsions Containing Catechin Subjected to Different pH Values

**DOI:** 10.3390/foods13020253

**Published:** 2024-01-12

**Authors:** Yuexin Li, Hui Wang, Yubo Zhao, Qian Chen, Xiufang Xia, Qian Liu, Baohua Kong

**Affiliations:** College of Food Science, Northeast Agricultural University, Harbin 150030, China; liyx980503@163.com (Y.L.); huiwang@neau.edu.cn (H.W.); zhaoybneau@163.com (Y.Z.); chenqianego7@126.com (Q.C.); xxfang524@163.com (X.X.); liuqian@neau.edu.cn (Q.L.)

**Keywords:** myofibrillar protein, diacylglycerol, pH, emulsifying property, oxidative stability

## Abstract

Myofibrillar protein–diacylglycerol emulsions containing catechin (MP-DAG-C) possess outstanding emulsifying property and oxidative stability. However, the effect of pH on MP-DAG-C emulsions should be revealed to provide possibilities for their application in practical meat products. Therefore, MP-DAG-C emulsions at different pH values were used in this study, in which lard, unpurified glycerolytic lard (UGL), and purified glycerolytic lard (PGL) were used as the oil phases. The results indicated that the emulsifying property of the UGL- and PGL-based emulsions increased compared to those of the lard-based emulsions (*p* < 0.05). The emulsifying activity and stability indices, absolute value of ζ-potential, and rheological characteristics increased with the increase in pH values (*p* < 0.05), with the droplets were smallest and distributed most uniformly at a pH of 6.5 compared to the other acidic environment (*p* < 0.05). The thiobarbituric acid substance and carbonyl content increased (*p* < 0.05), while the total sulfydryl content decreased (*p* < 0.05) during storage. However, there was no statistical difference between the oxidative stability of the MP-DAG-C emulsions with different pH values (*p* > 0.05). The results implied that the emulsifying property of MP-DAG-C emulsions increased with an increase in pH values. The oxidative stability of the MP-DAG-C emulsions at high pH values was improved by catechin.

## 1. Introduction

Myofibrillar proteins (MPs) exist in muscle fibers, which provide a variety of digestible nutrients to the human body [1]. Meanwhile, salt-soluble MPs provide outstanding texture and sensory properties due to their excellent emulsifying property, which determine the quality of meat products [2]. Fat is stabilized by the MPs in the system where protein and fat coexist [3]. Hydrophilic and hydrophobic groups of MPs are adsorbed at the oil–water interface to accelerate the dispersion of oil droplets [4]. Furthermore, changes in the structure and physicochemical properties of MPs can influence the interaction between the interfacial proteins and water phase to affect the stability of emulsions [3]. Consequently, MPs advance the stability of emulsions, and it is an urgent problem to maintain or improve the emulsifying property of MPs in the meat industry.

A higher content of hydroxyl groups in fats can improve the emulsifying properties by enhancing the interaction between fats and proteins. Diacylglycerol (DAG) is a type of fat with a hydroxyl group, which is provided with higher interfacial chemical properties and surface activity for stabilizing emulsions than triglyceride (TAG) [5]. Another study confirmed that the formation of gels containing MPs was facilitated by DAG [6]. Meanwhile, the expression of genes associated with lipolysis [7], fatty acid *β*-oxidation reactions [7], and the synthesis pathway of fatty acids [8] could be adjusted by DAG in the human body, resulting in the inhibition of the accumulation of fat to reduce the incidence of obesity. However, the low oxidative stability of DAG has been revealed [5], which possibly deteriorates the stability of emulsified meat products during storage. Lipid oxidation not only leads to a loss of vitamins and essential fatty acids, the deterioration of color, and a shortened shelf life and spoilage but also produces excessive free radicals to cause human DNA damage, heart disease, and cancer [9,10]. Equally, the oxidation of lipids triggers the adjacent protein molecules to cause oxidation, resulting in the intermolecular cross-linking, polymerization, or *β*-fragmentation of MPs and the deterioration of the stability of emulsions [11,12]. MPs also cause a loss in essential amino acids and the production of oxidation products, such as amino acid oxidation derivatives and protein carbonyls, which could have a negative impact on the nutrition and quality of emulsified meat products [13]. Therefore, the oxidation of fats and proteins must be inhibited to maintain the emulsifying property of emulsions containing MPs. Antioxidants are used to inhibit the oxidation of fats and proteins in meat products. Synthetic antioxidants could inhibit oxidative-induced adverse changes in meat products effectively, but they may have potential genotoxic effects [14]. In contrast, natural antioxidants are harmless and could inhibit the oxidation of proteins and fats [15]. Catechin is a natural hydrophilic polyphenol substance that serves as a metal chelator and free radical scavenger, inhibiting the formation of total protein carbon groups and protecting MPs from oxidative damage effectively [16]. The results of Zhao et al. [17] revealed that catechin could restrain the oxidation and stratification of emulsions. Meanwhile, the binding of catechin and proteins through non-covalent interactions alters the conformation of proteins, and promotes the unfolding of their structure, resulting in the tight combination of proteins and fats to stabilize the emulsion [18]. Furthermore, our previous study confirmed that the emulsifying property and oxidative stability of MP-DAG emulsions were advanced by catechin (MP-DAG-C).

Nevertheless, the emulsifying property of MPs is influenced by many conditions, such as the pH value, ionic strength, and temperature, which may cause a loss of water and fat to deteriorate the stability of emulsified meat products [19]. The pH value of different food systems vary between each other, which not only impacts the taste of the food but also influences the quality characteristics of the products [4]. Zhao et al. [6] investigated the rheological and physicochemical properties of MP-DAG gels at different pH values, which revealed that the pH value influenced the properties of MP-DAG gels significantly. The protonation state and α-carboxyl and α-amino terminal groups are altered at different pH values, which affects the dissociation of the amino and carboxyl groups in the MP molecules and influences the interaction between them, resulting in a decrease or increase in the emulsifying property of emulsions [20,21]. Furthermore, the oxidation of emulsions is impacted by the emulsifying property due to changes in the area in contact with the air [22]. Consequently, it is essential to evaluate the influence of pH on the emulsifying property and oxidative stability of MP-DAG-C emulsions based on the advantages of using MP-DAG-C emulsions in meat product processing applications. However, there are few studies on investigating the emulsifying property and oxidative stability of MP-DAG-C emulsions at different pH values.

In this study, MP-DAG-C emulsions subjected to different pH values were prepared, in which lard, unpurified glycerolytic lard (UGL), and purified glycerolytic lard (PGL) were used as the oil phases with different content of DAG, respectively. The emulsifying property was assessed by measuring the emulsifying activity (EAI), emulsifying stability index (ESI), particle size, absolute value of ζ-potential, and rheological properties, and applying confocal laser scanning microscopy (CLSM) analysis. Furthermore, the thiobarbituric-acid-reactive substance (TBARS), carbonyl, and total sulfydryl contents of the MP-DAG-C emulsions at different pH values were determined to evaluate their oxidative stability during storage. The present study offers a foundation for the study of MP-DAG-C emulsions and promotes their further application in emulsified meat products.

## 2. Materials and Methods

### 2.1. Materials

The pork backfat and lean meat were provided by a local market in Harbin (Harbin, Heilongjiang, China). The lipozyme RM was purchased from Novozymes A/S Co., Ltd. (Bagsvaerd, Denmark) and its lipase activity was 275 Interesterase Units Novo (IUN)/g. Furthermore, the (+)-catechin hydrate (Green tea, purity ≥ 98%) was obtained from Xi’an Entaiyuan Biotechnology Co., Ltd. (Xi’an, Shaanxi, China). All the reagents were of analytical grade without purification.

### 2.2. Extraction of the Myofibrillar Proteins

The MPs were extracted by using the methods of Xia et al. [23]. Crushed muscle was mixed with four times the amount of phosphate buffer solution (0.1 M NaCl, 2 mM MgCl_2_, 1 mM EGTA, and 10 mM phosphate buffer, pH 7.0). The obtained mixture was homogenized at 7000 rpm for 30 s using an IKA T 18 ULTRA-TURRAX homogenizer (IKA Werke GmbH & Co., Staufen, Germany) and centrifuged at 4200× *g* for 15 min. The above steps were duplicated three times to obtain pellets. The obtained pellets were washed with solution (0.1 M NaCl) and re-centrifuged three times to obtain the crude MPs. The crude MPs were kept at 4 °C and used within two days. The biuret reagent method was used to measure the concentration of the MPs.

### 2.3. Preparation of Diacylglycerol

The DAG was produced following the methods published previously [5]. Cut cubes of the backfat (1 cm × 1 cm × 1 cm) were heated at 120 °C to obtain the lard. The mixture of lard and glycerol was prepared at a molar ratio of 1:1 and lipase at a lard mass ratio of 4:100 was added into it. This mixture was subjected to magnetic stirring and incubated at 65 °C for 2 h. After that, the reacted mixture was transferred to 45 °C for 8 h at 500 rpm to gain the UGL. The UGL was conducted using two-step molecular distillation (SPE10, Haiyuan Biochemical Equipment Co., Ltd., Wuxi, China) to obtain the PGL. The content of DAG in the UGL and PGL was 62% and 82%, respectively.

### 2.4. Preparation of the Emulsions at Different pH Values

The MP-DAG-C emulsions were prepared by using the methods of Diao et al. [5] with slight modifications. The phosphate buffer solutions (0.6 M NaCl and 50 mM sodium phosphate) containing 20 μmol/g catechin with different pH values (5.0, 5.5, 6.0, and 6.5) were obtained by adjusting the volume ratio of the sodium phosphate. The obtained phosphate buffer solutions were used to dilute the MPs to obtain the MP solutions (10 mg/mL). Finally, 80% of the MP solutions and 20% of the obtained lipids were mixed and homogenized at 16,000 rpm for 60 s using an IKA T18 ULTRA-TURRAX homogenizer (IKA-Werke GmbH & Co., Staufen, Germany). The emulsifying property was evaluated immediately after preparation and the oxidative stability was measured after maintenance in the dark at 4 °C for 0 d, 3 d, 6 d, and 9 d, respectively.

### 2.5. Emulsifying Activity Indices and Emulsifying Stability Indices

The EAI and ESI were determined following the methods of Cho et al. [24]. After it was maintained for 10 min at 25 °C, 20 μL of each emulsion was extracted, and diluted 250 times with 1% sodium dodecyl sulfate (SDS) solution. Finally, the absorbance was analyzed at 500 nm, with 1% SDS solution used as the blank control group. The EAI and ESI were defined as followed:EAI (m^2^/g) = (2 × 2.303)/[*C* × (1 − *φ*)] × *A*_500_ × *D*(1)
ESI (%) = *A*_10_/*A*_0_ × 10^2^(2)
where *C* is the MP concentration before emulsification (g/mL), *φ* is the volume fraction of oil in emulsion (*v*/*v*), *D* is the dilution ratio of the emulsion, and *A*_0_ and *A*_10_ are the absorbances of the emulsion after standing for 0 min and 10 min, respectively.

### 2.6. Particle Size and Absolute Value of the ζ-Potential

The volume-weighted mean particle diameter (*d*_4,3_) was measured by using a laser particle size meter (SYNC, Michick, FL, USA). The continuous-phase refractive index was 1.330 and the dispersed-phase refractive index was 1.475 [5]. The absolute value of ζ-potential was determined by the nanoparticle degree and using a zeta potential analyzer (Nano ZS90, Malvern PANalytical Technology Co., Ltd, Malvern, UK).

### 2.7. Confocal Laser Scanning Microscopy

The CLSM was operated to capture the microstructure of MP-DAG-C emulsions following the methods of Zhang et al. [25]. The oil was dyed with 0.02% Nile Red dye and the protein was dyed with 0.1% Nile Blue A dye. The emulsion was diluted 1.2 times (1 mL) and dyed with 40 μL of the dyes for 30 min. Finally, the dyed emulsion’s microstructure was observed using confocal laser scanning microscopy (TCS SP8, Leica Co., Ltd, Heidelberg, Germany).

### 2.8. Rheological Property

The rheological property of the MP-DAG-C emulsions was determined using a modular rotary rheometer (HAAKE MARS 60, Thermo Fisher Scientific Technology Co., Ltd, Shanghai, China). The apparent viscosity was recorded, with the shear rate increasing from 0.01 to 10 s^−1^. The storage modulus (*G*′) and loss modulus (*G*″) were calculated within a shear rate range of 0.1–10 Hz.

### 2.9. Lipid Oxidation

The TBARS value was measured within 9 days following the methods reported by Mei et al. [26]. A mixture of the emulsion and thiobarbituric acid solution (1:4 *v*/*v*) was heated at 100 °C for 15 min. The reacted solution was cooled to 25 °C before centrifugation at 1800× *g* for 10 min. Eventually, the supernatant was taken for analysis at 412 nm using a TU-1800 spectrophotometer to calculate the TBARS value.

### 2.10. Carbonyl Content

The oxidation of the MPs was evaluated according to the carbonyl content following the methods presented by Oliver et al. [27]. The concentration of MPs in the MP-DAG-C emulsions was diluted to 2 mg/mL. A mixture of 1 mL DNPH (10-mM) and 1 mL of the emulsion (2 mg/mL) was reacted for 1 h and the reacted mixture was treated with 1 mL 20% TCA using centrifugation at 8500× *g* for 10 min to obtain a precipitate. The obtained precipitate was washed three times using 1 mL of a mixture solution of ethyl acetate and ethanol (1:1-*v*/*v*). The final precipitate was dissolved in 3 mL of 6 M guanidine hydrochloride solution at 37 °C for 15 min and centrifuged at 8500× *g* for 3 min to obtain a supernatant. The absorbance of the supernatant was analyzed at 370 nm.

### 2.11. Total Sulfydryl Content

The total sulfydryl content was measured following the previous methods of Benjakul et al. [28]. The concentration of MPs in the diluted MP-DAG-C emulsion was 2 mg/mL. The diluted emulsion and Tris glycine buffer (pH 8.0, 10 mM EDTA, 8 M urea, and 0.2 M Tris-HCl) at a ratio of 1:8 *v*/*v* were mixed and homogenized for 60 s. Subsequently, the mixture was centrifuged at 8500× *g* for 15 min to obtain a supernatant. The mixture of 0.5 mL DTNB (10 mmol/L) and 4.5 mL supernatant was kept for 30 min in the dark and centrifuged at 10,000× *g* for 15 min. The supernatant was analyzed at 412 nm using a TU-1800 spectrophotometer.

### 2.12. Statistical Analysis

All the samples were replicated in triplicate using three independent batches of samples and the data were analyzed using the Statistix 8.1 software package (Analytical Software, St. Paul, MN, USA). The results were performed as averages ± standard errors (SE) in this study. The identified significant differences between the group means at a level of *p* < 0.05 were evaluated using the Tukey procedure.

## 3. Results and Discussion

### 3.1. Emulsifying Activity Indices and Emulsifying Stability Indices

The EAI and ESI values of MP-DAG-C emulsions under different pH values are presented in Figure 1. It can be seen that the lard-based emulsions possessed the minimum EAI and ESI values (*p* < 0.05), while PGL-based emulsions exhibited the highest values (*p* < 0.05). On the one hand, the binding of the free hydroxyl group in DAG and water molecules leads to a tight combination of the oil and water phases to stabilize the emulsion [29]. On the other hand, fats are dispersed highly in the aqueous phase due to the two hydroxyl groups of monoglyceride presented in the UGL and PGL [30]. The findings of Long et al. [31] revealed that the stability of an oil-in-water emulsion was enhanced by peanut-oil-based DAG, which was in agreement with our results.

The EAI and ESI values of the MP-DAG-C emulsions increased significantly with the increase in pH value (*p* < 0.05). The results of Zhou et al. [32] revealed that the EAI value of the emulsions increased at higher pH values. The findings of Du et al. [33] also indicated that MP emulsions exhibited a stable state when the pH value was high. The results of the EAI and ESI were related to the following reasons closely. First, the formation of hydrogen bonds between proteins and waters and proteins and proteins is promoted by the exposure of the tyrosine residues on the surface of porous plasma proteins at a higher pH value [34]. Second, the muscle fibers are depolymerized sufficiently to improve the solubility of the MPs at a high pH value [35]. Third, the hydrophobic groups hidden within the molecule are exposed due to the unfolding of the MP structure, which could enhance the surface activity of the MPs, resulting in a tight packing of the MPs on the oil droplets to improve the emulsion stability at a higher pH value [4,36]. However, the electrostatic repulsion of MPs is weakened when the pH descends to pI (5.0–5.2), which is attributed to the weakening of hydrogen bonding for MPs with water, making it difficult to form intramolecular hydrogen bonds [33]. Consequently, the EAI and ESI values of the MP-DAG-C emulsions were lower at a pH of 5.0 and 5.5 than those at other pH values.

### 3.2. Particle Size and Absolute Value of ζ-Potential

The fat type and pH value changed the particle size and absolute value of ζ-potential for the MP-DAG-C emulsions (Figure 2). First, the *d*_4,3_ value of the UGL- and PGL-based emulsions was smaller (*p* < 0.05) and the absolute value of ζ-potential was higher (*p* < 0.05) compared to that of the lard-based emulsions. The small *d*_4,3_ value promotes the dispersion of droplets and a higher absolute value of ζ-potential produces strong electrostatic stability to prevent the merging of droplets [37,38]; thus, the UGL- and PGL-based emulsions possessed outstanding stability. The free hydroxyl group presented in DAG promotes the mixing of oil and water, which results in a small particle size for the emulsion [31]. The decrease in the *d*_4,3_ value increases the specific surface area of the particles, which improves the solubility of the MPs [39]. The high solubility of the MPs accelerates the diffusion of droplets, which promotes the homogenization of fats and proteins, resulting in a higher stability and the excellent emulsifying property of the MP-DAG-C emulsions [40].

The *d*_4,3_ value and absolute value of ζ-potential of the MP-DAG-C emulsions decreased (*p* < 0.05) and increased (*p* < 0.05) with the increase in pH value, respectively. These results indicated that a higher pH value was beneficial for maintaining the stability of the MP-DAG-C emulsions, which may be related to the following reasons. The pI of the MPs was 5.0–5.2 and the weakened electrostatic repulsion leads to the aggregation of proteins and the merging of oil droplets due to the reduction in charge when the pH is around pI, which has an adverse effect on the stability of the emulsion [4,41]. However, the protein has a more negative charge with the increase in the pH value, which leads to a flexible conformation and strong electrostatic repulsion force to promote the high solubility of the protein, resulting in a tight binding of protein and fat [4,42].

### 3.3. Microstructure

Figure 3 exhibits the CLSM results to characterize the microstructure of the MP-DAG-C emulsions at different pH values. The results showed that all the samples were oil-in-water emulsions and coated by a layer of MPs. The droplets of the UGL- and PGL-based emulsions were smaller and distributed more evenly compared to those of the lard-based emulsions, which indicated that DAG could maintain a stable state of emulsion. However, the lard-based emulsions exhibited the largest droplets, which accelerated the separation of the aqueous and oil phases in the emulsions [5]. It can be considered that the repulsive force between droplets is strengthened by the greater charges of emulsions containing DAG, which inhibits the aggregation of emulsion droplets effectively [5].

The size of the droplets reduced gradually, and their distribution became uniform when the pH value increased. The droplets of the MP-DAG-C emulsions tended to coalesce when the pH was 5.0 or 5.5, while there was a significant improvement at a pH of 6.0 and 6.5. These phenomena may be caused by the following factors: (a) The reduction in the charge density of the proteins and the repulsive strength of the protein interactions promote collisions between the protein molecules, resulting in the formation of larger droplets under a low pH values [43,44]. (b) The high solubility of the MPs at high pH values facilitates the exposure of buried hydrophobic groups and free thiol groups, as well as changes in the secondary structure, which are attributed to promoting the expansion of the interface and increasing the conformational flexibility, leading to enhancement of the binding of the water and oil phases [4,45]. (c) The loose and stretchy structure of the MPs at high pH values promotes the exposure of the hydrophobic and polar groups inside the protein peptide chain to bind with fat tightly, which improves the interaction of protein and fat to stabilize the emulsion [46].

### 3.4. Rheological Property

The rheological property of the MP-DAG-C emulsions is exhibited in Figure 4. It can be seen that the viscosity (Figure 4A), *G*′, and *G*″ values (Figure 4B) of the UGL- and PGL-emulsions were higher than those of the lard-based emulsions. The increase in the interfacial viscoelasticity of the MP-DAG-C emulsions hindered the movement of droplets, which prevented the coalescence of the emulsion to resist changes in the external environment [32,47]. The rheological property agreed with the results on the EAI, ESI, *d*_4,3_ value, and absolute value of ζ-potential and of the CLSM, which revealed that the stability of the emulsion was improved by DAG compared to TAG. The findings of Xu et al. [48] revealed that an emulsion containing DAG possessed a higher apparent viscosity compared to the others. It can be argued that DAG is adsorbed on the oil-water interface via hydrophilic and hydrophobic functional groups, which provides the ideal physicochemical properties to emulsions [49].

The viscoelasticity of the MP-DAG-C emulsions increased with an increase in pH value. The results indicated that the MP-DAG-C emulsions possessed outstanding emulsifying property at high pH values. The unfolding of the MPs and the exposure of non-polar amino acids at high pH values promote the interaction of the MPs to stabilize emulsions [50,51]. Meanwhile, the interfacial tension of the emulsion decreases with the increase in pH value, which indicates that MPs exhibit a heightened adsorption performance at higher pH values [4]. The close contact between particles at high pH values leads to strong interactions, which promotes the formation of a larger contact area and greater friction force, requiring greater stress to damage the structure [52]. Finally, the distribution of droplets of smaller particle sizes is more ordered and compact, resulting in an improvement in the viscosity and resistance to block flow [53].

### 3.5. Oxidation Stability of the Emulsions during Storage

The TBARS (Figure 5) and carbonyl content (Figure 6A) increased (*p* < 0.05), while the total sulfydryl content (Figure 6B) showed a downward trend (*p* < 0.05) during storage, indicating that the MP-DAG-C emulsions caused oxidation with a prolonged storage time. This may be attributed to the oxidation of proteins and fats during the storage of the MP-DAG-C emulsions. The oxidation of emulsions occurs at the oil-water interface commonly, which is attributed to free radials and transition metals being able to make contact with the hydroperoxides on the surface of the droplets [54].

In addition, the oxidation of the MPs and fats was relatively severe at high pH values, which indicated that the MP-DAG-C emulsions were more susceptible to oxidation with an increase in the pH value. The findings of Wang et al. [37] revealed that the oxidative stability of emulsions increased with a decrease in pH value. This result was related to the following reasons closely. First, the transition metal was iron, and a decrease in the aqueous solubility of iron at a high pH leads to the precipitation of insoluble iron on the surface of the emulsion droplets, which promotes the electrostatic attraction between the ionic oil droplets and cationic transition metals, resulting in an increase in the rate of lipid oxidation [55,56]. Second, an emulsion with smaller droplet sizes is more likely to accumulate primary oxidation products, which decreases the oxidative stability of the emulsion [22]. The buried hydrophobic groups and free thiol groups are exposed with an increase in the pH value to modify the conformation of proteins, which lead to accelerating the oxidation of sulfydryl [45]. The oxidative stability of the MP-DAG-C emulsions decreased slightly at high pH values, but there was no significant difference under different pH values on the same day, which revealed that catechin could reduce the oxidation of fats and proteins in MP-DAG-C emulsions at higher pH values. This may be attributed to the fact that the oxidation of proteins and fats in MP-DAG-C emulsions at high pH values is hindered by the presence of catechin [17]. The above results revealed that the oxidation of fats and proteins in the MP-DAG-C emulsions was delayed, which is profitable for prolonging the storage period of emulsified meat products.

## 4. Conclusions

This paper investigated the emulsifying property and oxidative stability of MP-DAG-C emulsions at different pH values. The emulsifying property of the MP-DAG-C emulsions was more outstanding than that of emulsions containing TAG. The results on the EAI, ESI, and rheology characterization for the MP-DAG-C emulsions were improved notably with an increase in the pH value. The droplets of the MP-DAG-C emulsions were smaller and distributed more uniformly at high pH values. The above results revealed that the emulsifying property of the MP-DAG-C emulsions increased with an increase in the pH value. The TBARS value and carbonyl content increased and the total sulfydryl content decreased with the prolongation of storage. However, there were no obvious changes in the oxidative stability of the MP-DAG-C emulsions at different pH values, which indicated that the presence of catechin improved the oxidative stability of the MP-DAG-C emulsions at high pH values. This study is beneficial for forecasting the application of DAG and catechin in practical meat products to promote the quality of emulsified meat products. However, other factors that influence the emulsifying property and oxidation stability of MP-DAG-C emulsions, as well as the application of DAG and catechin in practical meat products, should be investigated in the future.

## Figures and Tables

**Figure 1 foods-13-00253-f001:**
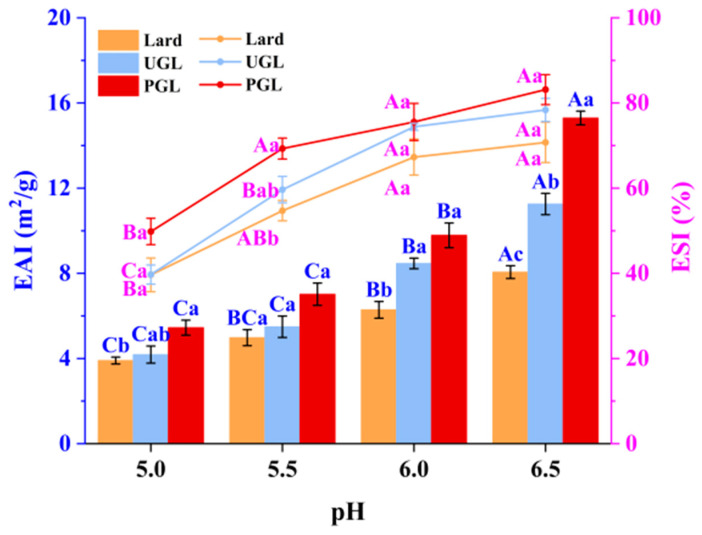
Emulsifying activity index (EAI) and emulsifying stability index (ESI) of MP-DAG-C emulsions at different pH values. ^A–C^ Means significant differences at different pH values for the same lipid-based emulsions (*p* < 0.05). ^a–c^ Means significant differences among different lipid-based emulsions at the same pH values (*p* < 0.05).

**Figure 2 foods-13-00253-f002:**
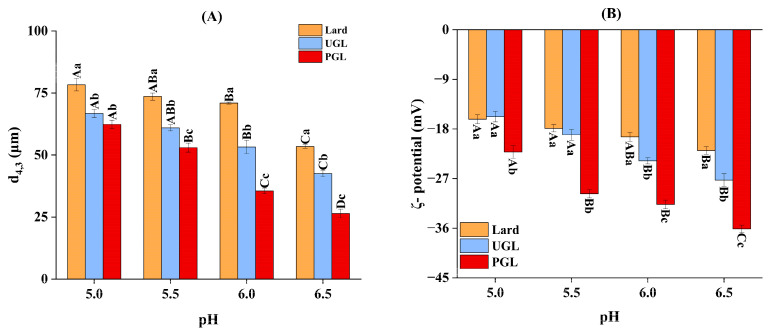
The volume-weighted mean diameter of particles (*d*_4,3_) (**A**) and *ζ*-potential (**B**) of MP-DAG-C emulsions at different pH values. ^A–D^ Means significant differences at different pH values for the same lipid-based emulsions (*p* < 0.05). ^a–c^ Means significant differences among different lipid-based emulsions at the same pH values (*p* < 0.05).

**Figure 3 foods-13-00253-f003:**
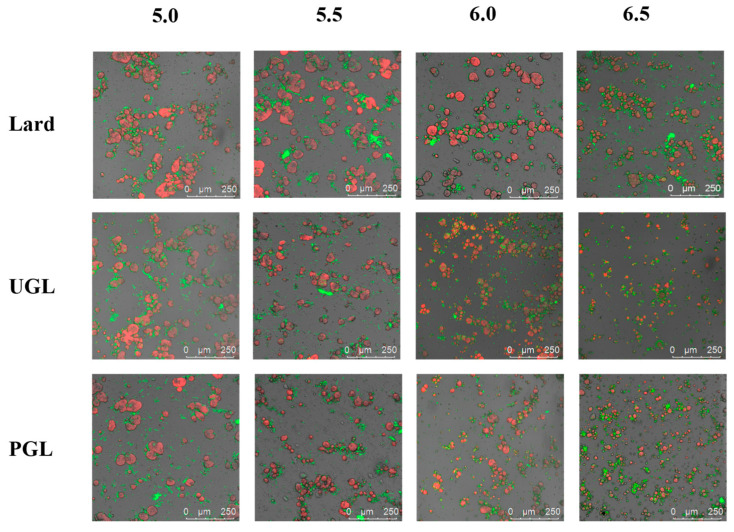
The confocal laser scanning microscopy of MP-DAG-C emulsions at different pH values. The pH values were 5.0, 5.5, 6.0, and 6.5.

**Figure 4 foods-13-00253-f004:**
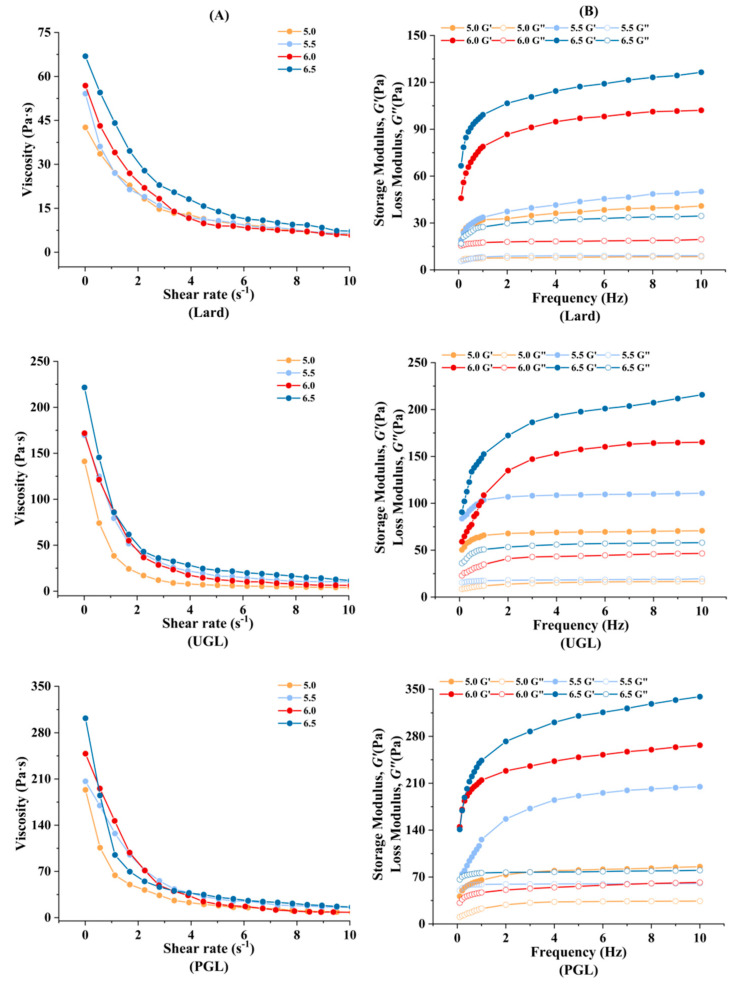
The apparent viscosity (**A**), storage modulus (*G*′), and loss modulus (*G*″) (**B**) of MP-DAG-C emulsions at different pH values.

**Figure 5 foods-13-00253-f005:**
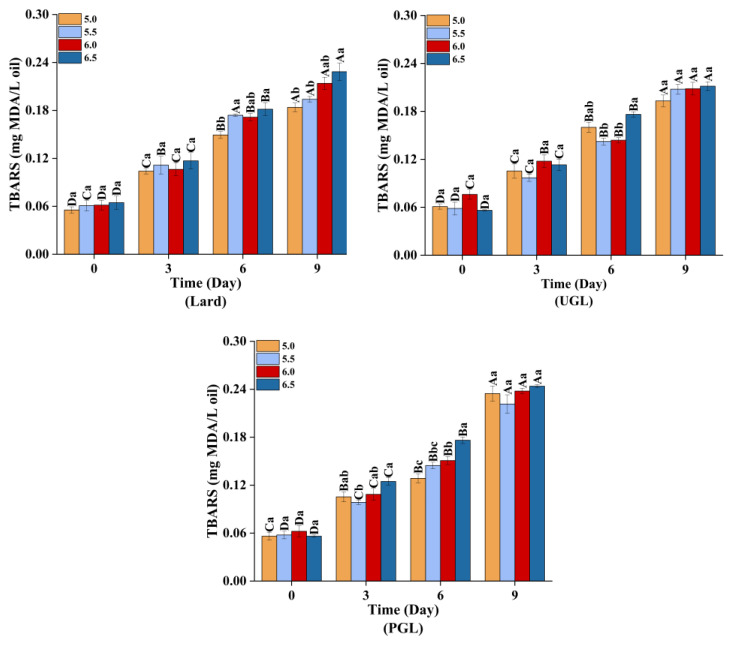
The TBARS values of MP-DAG-C emulsions at different pH values. ^A–D^ Means significant differences between different days at the same pH values (*p* < 0.05). ^a–c^ Means significant differences among different pH values on the same day (*p* < 0.05).

**Figure 6 foods-13-00253-f006:**
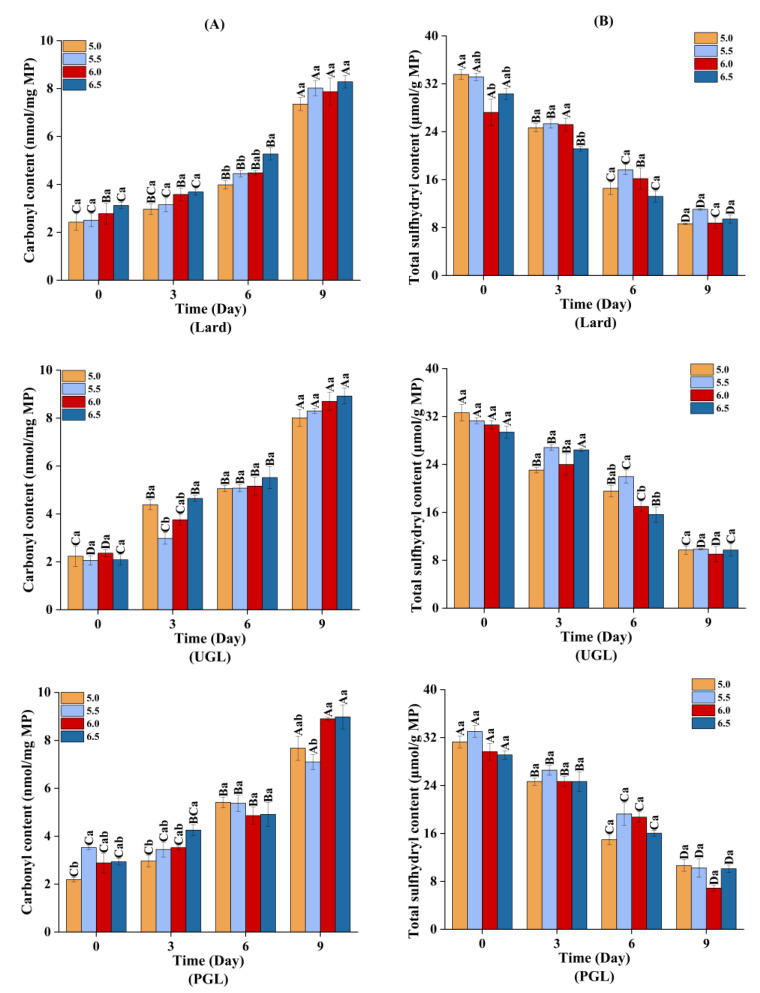
The carbonyl content (**A**) and total sulfydryl content (**B**) of MP-DAG-C emulsions at different pH values. ^A–D^ Means significant differences between different days at the same pH values (*p* < 0.05). ^a–c^ Means significant differences among different pH values on the same day (*p* < 0.05).

## Data Availability

The data presented in this study are available in the article.

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
