# Peer review of "Evaluation of the Emulsifying Property and Oxidative Stability of Myofibrillar Protein-Diacylglycerol Emulsions Containing Catechin Subjected to Different pH Values"

_foods, 2024, doi:10.3390/foods13020253_

Round 1

Reviewer 1 Report

Comments and Suggestions for Authors

Please see comments in attached files

Comments on the Quality of English Language

Document needs to be corrected for minor/moderate syntax and grammatical errors

Author Response

Response to Reviewer

Manuscript Number: foods-2816874

Title: Evaluation of emulsifying property and oxidative stability of myofibrillar protein-diacylglycerol emulsions containing catechin subjected to different pH values

Comments and Suggestions for Authors and Answers for Reviewer:

Q1: Line 19 Absolute ξ-potential value’ may be ‘ζ-potential??

A1: We are grateful for the reviewer to raise this important point. ‘Absolute ξ-potential value’ has been replaced as ‘the absolute value of ζ-potential’ in the revised manuscript (Line 19).

--------------------

Q2: Line 19 The grammar/syntax of rose as the increment of pH values should be revised.

A2: We are grateful for the reviewer’s suggestion. This sentence has been replaced as ‘... rheological characteristic increased with the increase of pH values (P < 0.05) ...’ in the revised manuscript (Line 19).

--------------------

Q3: Line 41 Emulsions do not have emulsifying effect. They are the result of the emulsifying properties of the emulsion constituents. Please rephrase.

Q3: We sincerely appreciate the valuable comments. We have revised in the revised manuscript (Line 41) as ‘A higher content of hydroxyl group in fats could improve the emulsifying properties through enhancing the interaction between fats and proteins’.

--------------------

Q4: Line 76 The grammar/syntax of ‘ is various’ should be revised.

A4: We are grateful for the reviewer’s suggestion. This sentence has been revised as ‘The pH value of different food systems is various from each other.’(Line 76).

--------------------

Q5: Line 87 Why catechin?

A5: Thank you for your comment and we are regret to make it confused for readers. 

Catechin as one of the effective antioxidant agent could restrain the oxidation and the stratification of oil and protein (Zhao, Jiao, Zhou, Lin, & Sun, 2018). Our previous study revealed that containing catechin (MP-DAG-C) emulsions exhibited excellent emulsifying property and oxidative stability. Thus, it is important to clarify the influence of some factors on these properties of MP-DAG-C emulsions to be applied in practical meat products, such as pH value and ionic strength. Consequently, this study evaluates the influence of pH on the emulsifying property and oxidative stability of MP-DAG-C emulsions based on the advantages of MP-DAG-C emulsions. Our previous statement actually caused ambiguity, which has been modified in the revised manuscript (Lines 64-73 and 86-90).

Zhao, M.; Jiao, M.; Zhou, F.; Lin, L.; Sun, W. Interaction of beta-conglycinin with catechin-impact on physical and oxidative stability of safflower oil-in-water emulsion. Food Chem. 2018, 268, 315–323. https://doi.org/10.1016/j.foodchem.2018.06.108 532.

--------------------

Q6: Line 92 these terms are not justified in the introduction enough to point out the significance of purified or unpurified glycerolytic lard etc.

A6: We apologize for the lack of such important information. The DAG content of UGL and PGL has been characterized according to Wang et al. (2011). The DAG content in lard, UGL, and PGL was 0%, 62%, and 82%, respectively, which has been added in the revised manuscript (Lines 93 and 127-128).

Wang, L.; Wang, Y.; Hu, C.; Cao, Q.; Yang, X.; Zhao, M. Preparation of diacylglycerol enriched oil from free fatty acids using lecitase ultra-catalyzed esterification, J. Am. Oil Chem. Soc. 2011, 88, 1557–1565. https://doi.org/10.1007/s11746-011-1821-0

--------------------

Q7: Line 132 The phosphate buffer had these pH values? Please clarify. How was the pH adjusted? eg NaOH, HCl?

A7: We are grateful for the reviewer to raising this important point. Different volume of acidic NaH2PO4 and the alkaline Na2HPO4 were used to prepare the phosphate buffer solutions with different pH values. The relevant description has been clarified in the revised manuscript (Line 133).

--------------------

Q8: Line 151 ξ-potential’ may be ‘ζ-potential??

A8: We are grateful for the reviewer to raise this important point. We have revised in the revised manuscript (Line 151).

--------------------

Q9: Line 238 ‘ξ-potential’ may be ‘ζ-potential??

A9: We sincerely appreciate the valuable comments. ‘ξ-potential value’ has been replaced as ‘the absolute value of ζ-potential’ in the revised manuscript (Line 238).

--------------------

Q10: Lines 263-269 What is the pI of the proteins? because depending on the PI value you can use or dismiss the explanation you give here...Usually (eg pH shift method) you use low or high pH (eg 3 or 10) to solubilize the proteins due to increased negative or positive charges.

A10: We appreciate the reviewer for your constructive suggestions. The detailed explanations have been added in the revised manuscript (Lines 263-269) as ‘The pI of MPs was 5.0-5.2, the weaken electrostatic repulsion leads to the aggregation of protein and merging of oil droplets due to the reduction of charges when pH was around pI, which adverses to the stability of emulsion (Li, Xu, Xu, Zeng, & Zhou, 2022; Owens, Griffin, Khouryieh, & Williams, 2018). However, the protein is accompanied by more negative charges with the increase of pH value, which leads to a flexible conformation and strong electrostatic repulsion force to promote the high solubility of protein, resulting in the tight binding of protein and fat (Li et al, 2022; Jiang et al., 2021)’.

Jiang, Y.; Zhu, Y.; Zheng, Y.; Liu, Z.; Zhong, Y.; Deng, Y.; Zhao, Y. Effects of salting-in/out-assisted extractions on structural, physicochemical and functional properties of Tenebrio molitor larvae protein isolates. Food Chem. 2021, 338, 128158. https://doi.org/10.1016/j.foodchem.2020.128158.

Li, Y.; Xu, Y.; Xu, X.; Zeng, X.; Zhou, G. Explore the mechanism of continuous cyclic glycation in affecting the stability of myofibrillar protein emulsion: The influence of pH. Food Res. Int. 2022, 161, 111834. https://doi.org/10.1016/j.foodres.2022.111834.

Owens, C.; Griffin, K.; Khouryieh, H.; Williams, K. Creaming and oxidative stability of fish oil-in-water emulsions stabilized by whey protein-xanthan-locust bean complexes: Impact of pH. Food Chem. 2018, 239, 314–322. https://doi.org/10.1016/j.foodchem.2017.06.096.

--------------------

Q11: Line 345 Which transition metals do you have in your preparations?

A11: Thank you for your comment. The transition metal was iron in this system, which has been added in the revised manuscript (Line 345).

--------------------

Q12: Lines 353-356 The fact that were no obvious differences, does not mean that they are stable. TBARS are increased at each measurement (days 0, 3, 6, 9). That means that they are not stable. eg for lard (and the others) on day 0 pH 5 has the lowest TBARS while day 9 they are significantly oxidized... Please be more clear and concise on what you are trying to say here.

A12: We are so sorry for our inaccurate description. The TBARS increased with the increase of storage days actually. However, the oxidative stability for MP-DAG-C emulsions decreased slightly with the increase of pH value at the same storage stage. And this illustrated that catechin could reduce the oxidation of fats and proteins in MP-DAG-C emulsions. This statement has been clarified in the revised manuscript (Lines 334-337 and 353-356).

--------------------

Comments on the Quality of English Language: Document needs to be corrected for minor/moderate syntax and grammatical errors.

Answers: We are grateful to the reviewer’s suggestion. The manuscript has been sent to an English native speaker to correct all the spelling and grammar errors to improve its language.

--------------------

Reviewer 2 Report

Comments and Suggestions for Authors

The manuscript is focused on the emulsifying effect and oxidative stability of myofibrillar protein diacylglycerol with catechin at different pH levels and it will be perfect after addressing the following minor concerns:

1. In the abstract, specify the problem that is going to be resolved in this research article.

2. Change the title to line number 238.

3. Provide the limitations of the study.

4. Many abbreviations are used in the whole manuscript. The author asked to provide a list of abbreviations at the end of the manuscript for clear understanding.

5. Line numbers 343–345: elaborate on the real application of your research.

Comments on the Quality of English Language

Minor English corrections are expected.

Author Response

Response to Reviewer

Manuscript Number: foods-2816874

Title: Evaluation of emulsifying property and oxidative stability of myofibrillar protein-diacylglycerol emulsions containing catechin subjected to different pH values

Comments and Suggestions for Authors and Answers for Reviewer:

Q1: In the abstract, specify the problem that is going to be resolved in this research article.

A1: We sincerely appreciate the valuable comments. The problem resolved in this research was to provide possibility for application of MP-DAG-C emulsions in practical meat products, which has been included in the abstract (Lines 13-15)

--------------------

Q2: Change the title to line number 238.

A2: Thanks for your suggestion and we have revised it as ‘3.3. Microstructure’ in the revised manuscript (Line 270).

--------------------

Q3: Provide the limitations of the study.

A3: We are grateful for the reviewer to raise this valuable point. This study is benefit for forecasting the application of DAG and catechin in practical meat products to promote the quality of emulsified meat products. However, other factors that influence the emulsifying property and oxidation stability of MP-DAG-C emulsions as well as the application of DAG and catechin in practical meat products would be investigated in the future. This limitation has been added in the conclusion section of the manuscript (Lines 379-383).

--------------------

Q4: Many abbreviations are used in the whole manuscript. The author asked to provide a list of abbreviations at the end of the manuscript for clear understanding.

A4: We sincerely appreciate the valuable comment. The list of abbreviations have been added at the end of the manuscript for clear understanding (Lines 386-391).

List of abbreviations:

Myofibrillar proteins (MPs); diacylglycerol (DAG); catechin (C); unpurified glycerolytic lard (UGL); purified glycerolytic lard (PGL); triglyceride (TAG); emulsifying activity index (EAI); emulsifying stability index (ESI); confocal laser scanning microscopy (CLSM); thiobarbituric acid-reactive substances (TBARS); sodium dodecyl sulfate (SDS); volume-weighted mean particle diameter (d4,3); storage modulus (G'); loss modulus (G").

--------------------

Q5: Line numbers 343-345: elaborate on the real application of your research.

A5: Thank you for your comment. We apologize for the confusion caused by the inaccurate description, and we have revised in the revised manuscript (Lines 379-380) as ‘This study is benefit for forecasting the application of DAG and catechin in practical meat products to promote the quality of emulsified meat products’.

--------------------

Comments on the Quality of English Language: Minor English corrections are expected.

Answers: We are grateful to the reviewer’s suggestion. The manuscript has been sent to an English native speaker to correct all the spelling and grammar errors to improve its language.

--------------------
